# A Peripheral Blood Signature of Increased Th1 and Myeloid Cells Combined with Serum Inflammatory Mediators Is Associated with Response to Abatacept in Rheumatoid Arthritis Patients

**DOI:** 10.3390/cells12242808

**Published:** 2023-12-09

**Authors:** Panagiota Goutakoli, Garyfalia Papadaki, Argyro Repa, Nestor Avgoustidis, Eleni Kalogiannaki, Irini Flouri, Antonios Bertsias, Jerome Zoidakis, Martina Samiotaki, George Bertsias, Maria Semitekolou, Panayotis Verginis, Prodromos Sidiropoulos

**Affiliations:** 1Laboratory of Rheumatology, Autoimmunity and Inflammation, Medical School, University of Crete, 71003 Heraklion, Greece; 2Rheumatology and Clinical Immunology, University Hospital of Heraklion, 71003 Heraklion, Greece; arrepa2002@yahoo.gr (A.R.); nesavgust@yahoo.gr (N.A.); iriald@yahoo.com (I.F.);; 3Department of Biotechnology, Biomedical Research Foundation, Academy of Athens, 11527 Athens, Greece; izoidakis@bioacademy.gr; 4Protein Chemistry Facility, Biomedical Sciences Research Center “Alexander Fleming”, 16672 Athens, Greece; samiotaki@fleming.gr; 5Institute of Molecular Biology and Biotechnology, Foundation for Research and Technology Hellas (FORTH), 70013 Heraklion, Greece; 6Laboratory of Cellular Immunology Division of Basic Research, Biomedical Research Foundation of the Academy of Athens, 11527 Athens, Greece; 7Laboratory of Immune Regulation and Tolerance, Division of Basic Sciences, Medical School, University of Crete, 71003 Heraklion, Greece

**Keywords:** rheumatoid arthritis, abatacept, CTLA4-Ig, biomarkers, immune cells, Th1, myeloid-derived suppressor cells, dendritic cells, proteomics

## Abstract

Abatacept (CTLA4-Ig)—a monoclonal antibody which restricts T cell activation—is an effective treatment for rheumatoid arthritis (RA). Nevertheless, only 50% of RA patients attain clinical responses, while predictors of response are rather limited. Herein, we aimed to investigate for early biomarkers of response to abatacept, based on a detailed immunological profiling of peripheral blood (PB) cells and serum proteins. We applied flow cytometry and proteomics analysis on PB immune cells and serum respectively, of RA patients starting abatacept as the first biologic agent. After 6 months of treatment, 34.5% of patients attained response. At baseline, Th1 and FoxP3+ T cell populations were positively correlated with tender joint counts (*p*-value = 0.047 and *p*-value = 0.022, respectively). Upon treatment, CTLA4-Ig effectively reduced the percentages of Th1 and Th17 only in responders (*p*-value = 0.0277 and *p*-value = 0.0042, respectively). Notably, baseline levels of Th1 and myeloid cell populations were significantly increased in PB of responders compared to non-responders (*p*-value = 0.009 and *p*-value = 0.03, respectively). Proteomics analysis revealed that several inflammatory mediators were present in serum of responders before therapy initiation and strikingly 10 amongst 303 serum proteins were associated with clinical responses. Finally, a composite index based on selected baseline cellular and proteomics’ analysis could predict response to abatacept with a high sensitivity (90%) and specificity (88.24%).

## 1. Introduction

Rheumatoid arthritis (RA) is a chronic autoimmune inflammatory disease, which may lead to articular bone and cartilage destruction, disability, and reduced life expectancy [1,2]. RA develops upon the aberrant activation of the immune system, mainly due to the failure of self-tolerance mechanisms [3]. T-helper (Th) lymphocytes have been demonstrated to play a central role in disease pathogenesis and progression through the release of cytokines and chemokines [4,5]. Both Th1 and Th17 cells are proven to contribute to RA inflammatory responses. On the other hand, CD4^+^ T-regulatory cells (FoxP3^+^ Tregs) as well as myeloid-derived suppressor cells (MDSCs) that play an immunoregulatory role and re-establish homeostasis are defective during active RA [6]. The restoration of immune regulation has been shown in RA patients upon effective treatment.

T-cell proliferation and the initiation of effector functions require intracellular signals. The first signal is mediated by the interaction between the antigen-presenting molecule HLA class II and the T-cell receptor (TCR), while the second signal is mainly mediated by an engagement of the cluster of differentiation (CD) 28 by co-stimulatory molecules, such as CD80/CD86, on dendritic cells (DCs) [7]. Cytotoxic T-lymphocyte antigen 4 (CTLA4) binds to the same ligands as CD28, with a higher affinity, and restricts T-cell activation [8]. CTLA4 expression is induced in all T cells transiently, after T-cell receptor activation, to regulate T-cell proliferation and the maintenance of tolerance [9]. Abatacept (CTLA4-Ig) is a recombinant fusion protein comprising the extracellular domain of human CTLA4 and a fragment of the Fc domain of human IgG1, which has been modified to prevent complement fixation [10]. Abatacept, like CTLA4, competes with CD28 for CD80 and CD86 binding to DCs, and thereby can be used to selectively modulate T-cell activation [11,12]. Abatacept has been approved for the treatment of RA, based on the results of an extensive clinical development program assessing its effectiveness and safety in different RA populations [13,14,15]. 

Biologic therapies (bDMARDs) have revolutionized the treatment of RA patients. Nevertheless, the data from registries and cohort studies have shown that, in clinical practice, 50–60% of RA patients starting bDMARD stop treatment due to inefficacy or toxicity in the long term, while a clinical response (remission or low disease activity) is unpredictable and is achieved by 20–40% of patients [16,17]. The development of predictive markers of response for everyday clinical practice is an important step in optimizing the treatment of autoimmune diseases. The available data for predictors of abatacept response are rather limited [18].

In this study, we propose that a detailed immunological profile performed early on treatment initiation with abatacept can differentiate between responders vs. non-responders. Moreover, a sera proteomics analysis revealed the signature of proteins associated with clinical responses. Immunological studies of peripheral blood (PB) and sera were performed at baseline, on the 3rd and 6th months after treatment initiation further characterized the biological effects of abatacept. Based on the abovementioned data, we propose a signature that can predict responses to abatacept.

## 2. Materials and Methods

### 2.1. Study Approval

This was an observational, prospective, single-center study of RA patients, starting treatment with abatacept, due to residual disease activity. All the patients were recruited by the Rheumatology and Clinical Immunology Clinic at the University Hospital in Heraklion, Crete. Decisions regarding the treatment scheme were made by the treating rheumatologist and according to the guidelines for RA treatment of the Hellenic Society of Rheumatology and EULAR guidelines. The study was approved by the Ethics Committee of the University Hospital of Heraklion (protocol number: 3601, 17 June 2015, Heraklion) and informed consent was obtained from all individuals prior to the sample collection stage.

### 2.2. Human Subjects 

Peripheral blood (PB) samples were obtained from RA individuals before the initiation of abatacept treatment and at 3 and 6 months upon treatment. Subjects with RA fulfilled the 2010 EULAR/ACR classification criteria at the Rheumatology and Clinical Immunology Clinic at the University Hospital of Heraklion [19]. At the time of sampling, all patients had an active disease state (DAS28 ≥ 3.2) and were naïve for biological therapy. Patients were followed clinically every 3 months and all immunological studies were performed at the indicated time points.

### 2.3. Peripheral Blood Mononuclear Cell Isolation

Heparinized blood (20 mL) was collected from individuals with RA. Peripheral blood mononuclear cells (PBMCs) were isolated on the Histopaque-1077 (Sigma-Aldrich, Merck KGaA, Darmstadt, Germany) density gradient. Briefly, the blood was diluted at a ratio of 1:1 with PBS and layered over Histopaque medium. Tubes were centrifuged at 400× *g* for 30 min at room temperature (RT) to obtain a PBMC layer. The PBMC layer was collected and the cells were washed with PBS. Erythrocytes were eliminated by hypotonic lysis (1 mL of ddH_2_O for 35 s and 1 mL of 1.8% NaCl). Then, the samples were processed for staining. Additionally, blood in appropriate serum-separator tubes was collected from individuals with RA. The tubes were centrifuged at 2500× *g* for 15 min at RT to obtain the serum.

### 2.4. Antibodies

Fluorescence-conjugated antibodies against humans with the following specificities were used to detect antigens by flow cytometry: CD4 (OKT4), IL-17 (BL168), CD127 (A019D5), CD25 (BC96), HLADR (L243), CD3 (OKT3), CD1α (HI149), CD14 (HCD14), CD15 (SSEA-1), and CD33 (WM53) from Biolegend; IFN-γ (B27) from BD Pharmingen; and FoxP3 (PCH101) from eBioscience.

### 2.5. Flow Cytometry

Single-cell suspensions were stained with fluorescence-conjugated antibodies and analyzed by flow cytometry. In brief, 3–5 × 10^6^ PBMCs were washed with FACS buffer (5% FBS in PBS), and the cells were then stained with the indicated antibodies for 20 min at 4 °C. Stained cells were washed twice with FACS buffer, resuspended and acquired on a FACS Calibur (BD Biosciences, Franklin Lakes, NJ, USA), and analyzed using FlowJo v10 software (Tree Star, Inc., Ashland, OR, USA).

### 2.6. Cellular Cytokine Expression Assays

Cytokine production was assessed through intracellular staining. In brief, to study Th cell populations (Th1, Th17), PBMCs were resuspended in a concentration of 3 × 10^6^ cells/mL in RPMI 1640 supplemented with 10% FBS, 100 IU/mL of penicillin, and 100 μg/mL of streptomycin cell culture media in 96-well flat-bottom plates and stimulated with PMA (50 ng/mL) and ionomycin (2 µg/mL) for 6 h in the presence of Brefeldin A (10 µg/mL) in a 5% CO_2_ atmosphere in a 37 °C incubator. Stimulation was stopped by washing the cells with cold PBS. Then, the cells were resuspended in the FACS buffer and stained with anti-CD4. For intracellular staining, the CD4-stained cells were fixed with a formaldehyde solution (4%) and permeabilized using saponin (10% *w*/*v*) and stained with conjugated antibodies against human IFN-γ and IL-17. For FoxP3 intracellular staining, the CD4-stained cells were permeabilized using the FoxP3 Staining Buffer Set (eBioscience, Carlsbad, CA, USA), according to the manufacturer’s instructions. Finally, the cells were washed with FACS buffer before their flow cytometric analysis.

### 2.7. Sample Preparation and Proteomics Analysis

The protein concentration of plasma samples was measured by the Bradford assay. An appropriate volume containing 200 μg of total protein per sample was processed with the filter-aided sample preparation (FASP) protocol [20]. Briefly, proteins were reduced with DTE (0.1 M), alkylated with iodoacetamide (0.05 M), and digested overnight by trypsin in 50 mM of NH_4_HCO_3_ at pH 8.5. The peptides originating from tryptic digestion were lyophilized and kept at −80 °C. The peptides were cleaned further using the Sp3 protocol for peptides. The samples were analyzed on a liquid chromatography tandem mass spectrometry (LC-MS/MS) setup consisting of a Dionex Ultimate 3000 nanoRSLC coupled with a Thermo Q Exactive HF-X Orbitrap mass spectrometer. Peptidic samples were directly injected and separated on a 25 cm-long analytical C18 column (PepSep, 1.9 μm^3^ beads, 75 µm ID) using a 1-hour long run, starting with a gradient of 7% Buffer B (0.1% formic acid in 80% acetonitrile) to 35% for 40 min and followed by an increase to 45% in 5 min and a 2nd increase to 99% in 0.5 min, and then kept constant for equilibration for 14.5 min. A full MS was acquired in profile mode using a Q Exactive HF-X Hybrid Quadrupole-Orbitrap mass spectrometer, operating in the scan range of 375–1400 *m*/*z* using a 120 K resolving power with an AGC of 3 × 10^6^ and maximum IT of 60 ms, followed by the data independent acquisition method using 8 windows (a total of 39 loop counts) each with a 15 K resolving power with an AGC of 3 × 10^5^ and max IT of 22 ms and normalized collision energy (NCE) of 26.

Orbitrap raw data were analyzed in DIA-NN 1.8.1 (Data-Independent Acquisition by Neural Networks) through searching against the Human Proteome (downloaded from Uniprot 20,583 protein entries, 8 November 2022) using the library free mode of the software, allowing up to two tryptic missed cleavages and a maximum of three variable modifications/peptides. A spectral library was created from the DIA runs and used to re-analyze them (double-search mode). The DIA-NN search was used with the oxidation of methionine residues with the acetylation of the protein N-termini set as the variable modification and the carbamidomethylation of cysteine residues as the fixed modification. The match between the runs feature was used for all analyses and the output (precursor) was filtered at 0.01 FDR, and finally the protein inference was performed on the level of genes using only proteotypic peptides.

Perseus software (version1.6.15.0) was used for the statistical analysis of the raw data. The baseline proteomic dataset of the cohort was grouped based on the active swollen and tender joints count after 6 months of abatacept therapy. The groupings were responders and non-responders. The dataset was Log2 transformed and filtered based on a threshold of at least 70% valid values in at least 1 of the 3 groups. The groupings were statistically evaluated using an ANOVA test. The final list of significantly altered proteins contained 10 proteins with *p*-values lower than 0.05.

The mass spectrometry proteomics data were deposited in the ProteomeXchange Consortium via the PRIDE partner repository with the dataset identifier PXD046112 [21].

### 2.8. Statistics

An unpaired *t*-test, one-way ANOVA, or Wilcoxon matched-pairs test were applied in all the experimental settings. All the data were analyzed using GraphPad Prism v8 software. Differences were considered statistically significant at a *p*-value < 0.05.

## 3. Results

### 3.1. Patients’ Characteristics at Baseline and the Effect of Treatment

We enrolled 29 RA patients who required the initiation of bDMARD due to having an active disease (median DAS28-ESR = 5.48). The demographics and clinical characteristics at baseline and at 6 months of abatacept therapy of RA patients are presented in Table 1. All patients received abatacept as the first biologic therapy. A total of 89.7% of patients were on concomitant treatment with MTX (mean dose: 17.9 mg/week; range: 10–25 mg/week), while 34.5% were on steroids. Patients were followed clinically every 3 months and disease activity was documented based on the active swollen and tender joints count. Patients with ≤2 swollen joints at 6 months were categorized as responders; the rest were referred to as non-responders. We also applied the EULAR response criteria to categorize clinical responses (i.e., good, moderate, or no response). After 6 months of therapy, 10 patients were responders (≤2 swollen joints), while based on the EULAR criteria, 7 patients were good and 10 moderate responders. PB and serum were collected before treatment, and then at 3 and 6 months after the start of abatacept treatment, detailed immunological profiling as well as proteomic analyses were performed (Figure 1).

### 3.2. High Disease Activity of RA Is Reflected in Serum Proteome Levels

The expression pattern of serum proteins was altered in disease conditions, and since it is known that proteins play an important role in the pathogenesis of RA, we performed proteomics to characterize the immunological profiles of RA patients. The serum proteomic analysis was filtered based on 50% of total valid values/proteins and the dataset that was generated, through a z-scoring normalization, identified 303 unique proteins (Figure 1a). Specifically, the enrichment analysis showed that these proteins participated in pathways, such as inflammatory responses, metabolic processes, complement systems, macrophage markers, oxidative damage (Figure 1b), highlighting the active inflammation on the sera of the RA patients enrolled in the study.

### 3.3. CTLA4-Ig Decreases the Proportion of CD4^+^ T Cells in RA Patients 

Abatacept binds to co-stimulatory molecules on DCs and inhibits interactions with T cells, thus restricting T-cell activation. However, the impact of abatacept on cell survival and proliferation remains elusive. So, firstly we aimed to explore the effect of abatacept on the frequency of the PB cell populations of treated patients. Since abatacept is not a cell-specific depleting antibody, we studied cell populations that triggered inflammatory responses in RA, such as Th1 (CD4^+^IFN-γ^+^) and Th17 (CD4^+^IL-17^+^), or played a regulatory role in immune responses, like FoxP3^+^ T cells (CD4^+^CD127^-^CD25^+^FoxP3^+^ T cells) and MDSCs (CD14^+^HLADR^int/−^CD15^+^CD33^+^ cells), as well as DCs (CD3^-^HLADR^+^CD1α^+^ cells) that provided a crucial link between innate and adaptive immune responses (Figure 2a). 

Thus, the proportions of CD4^+^ T-cell subsets at baseline and at 6 months of abatacept treatment were evaluated. Notably, both pathogenic, Th1 and Th17, and regulatory FoxP3^+^ T cells were significantly reduced in CTLA4-treated RA patients. Furthermore, we studied the impact of CTLA4-Ig on myeloid cell populations, and we determined no significant differences in the levels of MDSCs and DCs after 6 months of abatacept treatment in the total cohort (Figure 2b). 

### 3.4. Disease Activity Is Positively Correlated with the Proportion of CD4^+^ T-Cell Subsets

Then, we aimed to investigate the association of baseline levels of lymphocytic and myeloid cell populations with the clinical characteristics of RA patients, such as seropositivity, ESR, CRP, joint counts, patient VAS, and DAS28-ESR. We concluded that high levels of Th1 and FoxP3^+^ T cells were positively associated with disease activity based on the Tender Joint 28 count (*p*-values = 0.047 and 0.022, respectively), while the proportion of MDSCs were elevated in seropositive RA patients (*p*-value = 0.05) (Table 2).

### 3.5. Increased Baseline Proportion of Th1 Cells Is Associated with the Response to Abatacept Therapy

Aiming to assess whether baseline immune cell populations may predict clinical responses, we compared cell subsets at baseline between responders and non-responders at 6 months. Since CD4^+^ T cells play a central role in the pathogenesis of RA, we determined their frequency, and we observed lower levels of CD4^+^ T cells in the responders compared to non-responders (Figure 3a). However, the levels of Th1 cells at baseline were significantly increased in the responders compared to non-responders at 6 months of abatacept treatment (Figure 3b). In accordance with this result, the frequency of Th17 cells at baseline tended to be elevated in responders, but significant differences at baseline according to the clinical response at 6 months were not observed (Figure 3c). Furthermore, the proportion of FoxP3^+^ T cells between the responders and non-responders was comparable (Figure 3d). So, our findings suggest that the value of the pathogenic Th1-cell population is associated with the response rate at 6 months following abatacept therapy.

Then, we studied the changes in the proportions of CD4^+^ T-cell subsets at baseline, 3 and 6 months after abatacept treatment according to the clinical response. We observed that pathogenic Th cells were reduced only in responders, while the levels of FoxP3^+^ T cells were not affected upon abatacept therapy both in the responders and non-responders (Figure 3e–g).

### 3.6. Baseline Myeloid Cells Are Elevated in Responders

Furthermore, we investigated the role of MDSCs as potent predictors of clinical response, as well as the effect of abatacept on the proportion of MDSCs separately on the two groups of patients, and we found that the percentage of regulatory MDSCs at baseline was significantly higher in responders compared to those with an active disease at 6 months (Figure 4a). Interestingly, we showed that MDSCs were reduced at 6 months only in the responders (Figure 4b).

Abatacept restricts T-cell activation by blocking the interaction of CD80/CD86 on DCs with CD28 on T cells. Thus, we assessed whether the frequency of DCs was associated with a response to therapy. Interestingly, we found that the frequency of DCs was elevated in responders as compared to non-responders (Figure 4c). Notably, the levels of DCs decreased in the PB of responders during abatacept therapy, but not in non-responders, showing that the reduced DC population was accompanied by lower disease activity (Figure 4d,e). Overall, these results indicate that myeloid cell populations, MDSCs, and DCs in the PB of RA patients upon the initiation of abatacept treatment are associated with a clinical response after 6 months.

### 3.7. Inflammatory Mediators Are Present in the Serum of Responders before Therapy Initiation

Furthermore, we analyzed serum proteome to identify the proteins that were associated with a clinical response. A statistical comparison identified 10 differentially expressed proteins at baseline in responders compared to non-responders (*p*-value < 0.05) (Figure 5a). Representative processes are presented in Table 3 with information on the protein function and expression ratio of responders vs. non-responders. These deregulated proteins were involved in processes, such as a response to stimulus, activation of an immune response, metabolic processes, cell differentiation, and cytoskeleton organization. Notably, most of the differentially expressed proteins were implicated in immune response pathways [22,23,24] (Figure 5b). Specifically, we found that elevated levels of CSF1R, GC, ADIPOQ, TNXB, and CTBS were associated with a response to abatacept. Moreover, we concluded that high levels of proteins, such as immunoglobulins, apolipoproteins, and the complement system, were negatively correlated with a response to therapy. These findings suggest that the serum proteome analysis of RA patients in combination with immune profiling offer insights into responses to abatacept therapy.

### 3.8. A Composite Cellular and Proteomic Index Predicts the Response to Abatacept

Finally, to assess whether combining the cellular and proteomic data could better predict responses, we formulated a composite index. This was based on the following values: (1) Values higher than the highest quartile (75%) of the 5 highest-expressed proteins. (2) Values of lowest quartile (25%) of the 5 proteins with lower baseline values. The sum of each one of the true-high or -low proteins was the “protein” score (range: 0–10). (3) Values higher than the highest quartile (75%) of patients with Th1, MDSCs, and DCs. The sum of each one of the true-high cellular values was the “cellular” score (range: 0–3). We finally combined the three abovementioned values in a “compo index” (range: 0–13). We denoted a high “compo index” with a value of ≥5. The performance of a score ≥5 of the “compo index” to predict responses was high (AUC = 0.93, 95% CI 0.83 to 1), with 90% sensitivity and 88.24% specificity, and an odds therapy response of 67.5 (*p*-value < 0.0001) (Appendix A).

## 4. Discussion

Although several trials proved the clinical effectiveness of CTLA4-Ig, only 40–50% of patients attained desired clinical responses. Responses to abatacept as well as to most of the biologic agents used to treat RA are unpredictable, since no clinically reliable prognostic markers are available to assist individual agent selections. In this study, we investigated whether the PB immunological profiling and serum proteome of RA patients could be used as biomarkers to predict patients’ clinical responses to abatacept treatment. We observed a strong association of high levels of Th1, MDSCs, and DCs with a better response to abatacept treatment at 6 months. Moreover, baseline levels of 10 out of 303 proteins in peripheral serum showed a differential expression according to clinical responses at 6 months. Interestingly, a composite index based on the abovementioned cellular and protein markers showed a high discriminative ability to predict a response to abatacept therapy.

Several animal and human studies have shown that both Th1 and Th17 effector T cells contribute to RA pathogenesis. Previous studies have shown that IFN-γ-producing CD4^+^ cells are present in RA synovium [25,26], while more recent synovium RNA-seq analysis data support the presence of IFN-γ in early RA patients [27]. Mostly, animal studies strongly support the contribution of Th17 cells to synovial inflammation [28,29,30]. On the contrary, FoxP3^+^ Tregs detected in RA tissues have a compromised function, highlighting that the imbalance of pathogenic effector T cells and regulatory T cells is an essential factor for the development of RA [6]. Given that CTLA4 is expressed and functionally regulates the abovementioned effector and regulatory T cells, we question whether these subpopulations may affect responses to abatacept. We found that responders had higher numbers of CD4^+^IFN-γ^+^ cells and a trend for higher CD4^+^IL-17^+^ T cells at baseline, compared to patients resistant to abatacept. These findings indirectly support the idea that the therapeutic effect of abatacept in in the PB of RA patients can be partially attributed to a direct inhibitory effect on these effector T cells. Although several studies assessed for soluble biomarkers to predict responses to abatacept, only a few assessed whether immune cells in PB could be associated with clinical responses.

A microarray analysis of the PB of RA patients before abatacept therapy revealed that a high expression of type-I IFN-related genes was associated with a clinical response to abatacept [31]. In our study, we did not assess for type-I IFN levels; however, we showed that IFN-γ-producing CD4^+^ T cells were strongly associated with clinical responses to abatacept in RA patients. Inamo et al. showed that elevated levels of Th17 and Treg cells could be associated with clinical responses to abatacept in seropositive early RA patients [32]. However, in our cohort, seropositivity was not a co-factor—together with CD4^+^IFN-γ^+^ levels—which affected clinical responses. The differences in the results for the abovementioned studies can be attributed to the clinical and genetic differences between the patients studied, the stage of RA (early vs. established), and the methods applied to investigate peripheral immune cells.

Additionally, we found that CTLA4-Ig significantly reduced PB pathogenic CD4^+^ T cells (Figure 2b); nevertheless, the effect was more significant, irrespective of the clinical responses to Th17 cells (Figure 3f), while the reduction in CD4^+^IFN-γ^+^ was significant only for clinical responders (Figure 3e). This finding is in accordance with the data showing that abatacept decrease circulating CD8^+^CD28^−^ T cells and other effector T cells [33], as well as CD25^+^ Tregs [32]. Nevertheless, the idea that abatacept mediates these effects has not been investigated.

MDSCs are a heterogeneous population of myeloid cells that have a regulatory role through an increase in regulatory cells and the inhibition of pathogenic T cells [34]. Although some reports have shown that MDSCs can aggravate inflammatory arthritis in mice [35,36,37], in most of studies, the regulatory role of MDCSs has been observed. MDSCs played a crucial role in the suppression of collagen-induced arthritis (CIA) in a mouse model by inhibiting CD4^+^ T-cell proliferation, the differentiation into Th17 cells, and cytokine production [38]. In vitro studies have indicated that MDSCs regulate Th17/Treg cells and decrease Th1 and Th17 cells, while Tregs are increased via IL-10, ameliorating inflammatory arthritis in vivo [39,40]. Furthermore, it has been found that RA patients with high disease activity have an increased frequency of PB MDSCs that are also detectable in the synovial tissues of patients [41,42,43]. However, the impact of abatacept on MDSCs as well as any association between the proportion of MDSCs and therapeutic response are unknown. In this study, although we observed no depleting effect in the MDSC population after 6 months of abatacept therapy (Figure 2b and Figure 4b), interestingly, the baseline levels of MDSCs were positively associated with clinical responses at 6 months (Figure 4a). It has been shown that MDSCs express low levels of CD80 [37], but whether any effect of CTLA4-Ig on these cells can be mediated through reverse signaling in this mostly suppressive population is not known.

DCs are at the crossroads of immune responses through their crucial role in the antigen presentation process and their dual role for both T-cell activation or inhibition. RA DCs have an increased capability to recruit macrophages, neutrophils, and monocytes due to their enhanced secretion of chemokines, CXCL8 and CCL3, leading to exacerbated inflammation [44]. Moreover, DCs exist in high concentrations in RA synovial joint tissues and secrete increased amounts of pro-inflammatory cytokines IL-12 and IL-23, inducing the generation of Th17 cells [45]. Synovial DCs also secrete high levels of chemokines, CCL17, CXCL9, and CXCL10, attracting effector T cells, which increase local inflammation, underlying their role in the initiation of joint inflammation [46]. There are clinical trials using ex vivo manipulated tolerogenic DCs in patients with RA; however, they have limited efficacy [47,48]. Regarding abatacept, it is well studied that CTLA4-Ig induces its immunoregulatory function by binding to co-stimulatory molecules on DCs mediating a reverse signaling effect [49,50]. Our findings suggest that a higher frequency of DCs upon abatacept initiation is positively correlated with a response to abatacept therapy (Figure 4c). This finding further supports the idea that the anti-inflammatory effect on RA patients may at least be partially mediated through the abovedescribed reverse signaling of DCs. Interestingly, our finding that DCs’ numbers are stable during 6 months of therapy (Figure 2b and Figure 4e) supports the concept of a continuous immunomodulatory effect through CTLA4-Ig on DCs, which can be of clinical significance.

Herein, we performed a serum proteomic analysis before abatacept treatment and detected proteins considered to contribute to the inflammatory response in RA. Among the 303 proteins assessed, we concluded with 10 differentially expressed proteins regarding the response to therapy after 6 months, and the majority of them were known to contribute to the pathogenesis of disease.

Interestingly, most of the proteins positively correlated with response had a pro-inflammatory role. Thus, CSF1R, which binds to colony-stimulating factor-1 (CSF-1) and promotes the proliferation and differentiation of myeloid cells, was found to be expressed in the synovial tissues of RA and PA patients [51]. Pharmacological inhibitors of CSF1R reduced the inflammatory activation of RA synovial tissue and severity of experimental arthritis [52]. ADIPOQ, which is a collagen-like protein, shown to have a pro-inflammatory role in RA through the secretion of inflammatory mediators, was found to be positively correlated with response [53]. Interestingly, high levels of adiponectin were detected in the serum and synovial fluid of RA patients [54]. TNXB is a component of the synovial membrane, and immunofluorescence labeling in the synovium of RA patients shows a significant expression, suggesting that inflammatory mediators may increase TNXB production [55]. Finally, CTBS, which is a lysosomal glycosidase and participates in the degradation of glycoproteins, was also increased in the serum of the responders. A previous report revealed increased levels of CTBS in RA patients after tocilizumab treatment [56].

Furthermore, five of the proteins detected in our analysis were shown to have a negative correlation with a response to abatacept. ApoC3 is a regulator of lipoproteins and serves as a link between insulin resistance (IR) and beta-cell dysfunction that are present in RA patients. Notably, high ApoC3 serum levels have a positive and significant association with disease activity in RA patients [57]. IGHV2-70D and IGHD are immunoglobulin heavy-chain (IGH) proteins that influence the B-cell-receptor altering response to infections. A variety of IGH proteins, as well as polymorphisms in IGH loci, are associated with autoimmune diseases, like RA and systemic lupus erythematosus (SLE) [58,59]. Regarding the properdin complement system, it was found that the neutralization of properdin played a protective role in the development arthritis in mice models [60]. Herein, we found that complement factor properdin (CFP) was abundant in the serum of non-responders to abatacept.

A limitation of our study was the rather low number of patients recruited. Nevertheless, in most of the studies assessing predictors for abatacept response, a rather comparable number of patients has been studied, ranging from 35 up to 59 [31,61,62]. An additional limitation was the lack of control-group patients treated with another biologic agent, in order to address the issue of the specificity of the results for abatacept as compared to other biologic agents. Finally, the lack of validation of the findings in another independent cohort of RA patients starting abatacept can be considered as a limitation.

Although abatacept is an effective treatment for RA, only 30–40% of patients responded to therapy and individual responses were unpredictable. The research focuses on the area of biomarkers to predict responses to biological agents at the individual level, but reliable and clinically applicable biomarkers are not available. Herein, we formulated a “composite index” based on the expression levels of the cellular signature and proteins found to be associated with clinical responses. The index was observed to have a high discriminative value to predict responses (AUC = 0.93, 95% CI 0.83 to 1.00), with 90% sensitivity and 88.24% specificity for clinical responders (Appendix A). If these data are confirmed in a larger cohort, they would be of clinical value to assist individual treatment choices and further optimize RA treatment.

## Data Availability

The data that support the findings of this study are available from the corresponding author upon reasonable request. Proteomics data are available via ProteomeXchange with identifier PXD046112.

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
