# Peer review of "A Peripheral Blood Signature of Increased Th1 and Myeloid Cells Combined with Serum Inflammatory Mediators Is Associated with Response to Abatacept in Rheumatoid Arthritis Patients"

_cells, 2023, doi:10.3390/cells12242808_

Round 1

Reviewer 1 Report

Comments and Suggestions for Authors

Authors in this study propose that a detailed immunological profile early at treatment initiation with abatacept, could differentiate responders vs non-responders. Finally they propose a signature which could predict response to abatacept. 

Introduction

-Correct the sentence: by both.the T cell receptor a co-receptor, CD28. 

Patient selection and characteristics

-       RA patients, starting treatment with abatacept, due to residual disease activity. It surprised me that low inflammatory markers and 50% of them were seronegative (both RF and anty-ccp) ? Can you support us with data from the time of diagnosis?

-       Could you give data of methotrexate dose?

-       Is disease duration reported in months (see table 1)

Results

-       Authors presented heat map of Proteomics analysis of RA patients sera at baseline. Could you present results for 6 mo time point?

-       Authors declare that ended up that high levels of Th1 and FoxP3T cells were positively associated with disease activity based on Tender Joint 28 count, while proportion of MDSCs were elevated in seropositive RA patients (Table 2) I do not see this in the table 2.

-       Title: Increased baseline levels of Th1 cells are associated with response to abatacept therapy. I would suggest use levels but numbers or proportion or frequency.

Discussion

-       Paragraph of limitation of the studies is missing.

-       Explain in what way your idea seems improve current clinical practice and is better than classical profile based on ESR, CRP, seropsitivity and SJC in prediction of the response

Reviewer 2 Report

Comments and Suggestions for Authors

The study aims to develop predictive markers response after using of 

 abatacept (CTLA4-Ig) in rheumatoid arthritis patients. They compared the immunological phenotypes of inflammatory cell, and serum proteins signature between baseline, 3 months, and 6 months after treatment. 

Minor comments:

1-   Abstract: I think the abstract should be written in a better way such as, stating the main aim of the study which is development predictive markers investigations of treated patients.

2-   The figures: some figures need to be more clarified (more resolution) especially Fig 5a.

3-   I think adding of abbreviations list will improve the article because it has a lot of uncommon acronyms especially the proteins.

Overall the study provide a good way to predict one of the biological treatment response in one of most common autoimmune disease, however, the sample size is small. 
